# Molecular Mechanisms in Early Diabetic Kidney Disease: Glomerular Endothelial Cell Dysfunction

**DOI:** 10.3390/ijms21249456

**Published:** 2020-12-11

**Authors:** Emelie Lassén, Ilse S. Daehn

**Affiliations:** Division of Nephrology, Department of Medicine, Icahn School of Medicine at Mount Sinai, New York, NY 10029, USA; sandra.lassen@mssm.edu

**Keywords:** diabetic kidney disease, glomerular endothelial cells, ROS

## Abstract

Diabetic kidney disease (DKD) is the leading cause of end-stage renal disease (ESRD), with prevalence increasing at an alarming rate worldwide and today, there are no known cures. The pathogenesis of DKD is complex, influenced by genetics and the environment. However, the underlying molecular mechanisms that contribute to DKD risk in about one-third of diabetics are still poorly understood. The early stage of DKD is characterized by glomerular hyperfiltration, hypertrophy, podocyte injury and depletion. Recent evidence of glomerular endothelial cell injury at the early stage of DKD has been suggested to be critical in the pathological process and has highlighted the importance of glomerular intercellular crosstalk. A potential mechanism may include reactive oxygen species (ROS), which play a direct role in diabetes and its complications. In this review, we discuss different cellular sources of ROS in diabetes and a new emerging paradigm of endothelial cell dysfunction as a key event in the pathogenesis of DKD.

## 1. Introduction

Diabetes is the leading primary cause of end-stage renal disease (ESRD) [1] and is increasing in prevalence at an alarming rate worldwide [2]. The economic burden is substantial, as the costs of diabetes management in 2019 were estimated to be approximately US $760 billion and expected to increase to US $845 billion by 2045, the majority of which will be spent on the treatment of diabetic complications [3]. Therefore, new therapeutic approaches to prevent the progression of DKD are an urgent unmet medical need and subject to intense efforts by the medical research community and pharmaceutical industry.

Approximately 20–40% of diabetic patients develop DKD even with comparable blood glucose levels. Clinical diagnosis of DKD relies on the presence of persistent albuminuria, reduced estimated glomerular filtration rate (eGFR) and the presence of other diabetes-related complications such as retinopathy [4,5]. Metabolic dysregulation, including hyperglycemia and dyslipidemia, drives the early pathological changes in DKD. The glomerulus is the primary site of diabetic injury in the kidney, and hallmarks of progressive DKD include glomerular hyperfiltration and alterations in the production and composition of the extracellular matrix, leading to mesangial matrix expansion and increased thickness of the glomerular basement membrane (GBM), hence decreasing glomerular filtration surface area [6]. These changes are all early predictors of DKD progression [6,7,8]. Furthermore, the degree of podocyte damage and depletion also correlates closely with the severity of the disease, and this process is proceeded by albuminuria, glomerular sclerosis and eventual declining renal function [9,10,11].

Although hyperglycemia and hypertension are known to drive the onset and progression of DKD, intensive glycemic control has only had modest effects and fails to stop DKD progression to ESRD and death [1,12]. Although new therapies are emerging today, the absolute risk of renal and cardiovascular morbidity and mortality, as well as the need for renal replacement therapy, remains high. Therefore, elucidation of the mechanisms underlying DKD and the development of new and more effective approaches to the prevention of renal dysfunction and treatment requires a better understanding of disease mechanisms. The next sections will present up to date literature on the emerging evidence of endothelial cell dysfunction and the potential mechanisms involved in early DKD pathogenesis.

## 2. The Diabetic Milieu Affects Structure and Function of the Interconnected Glomerular Filtration Barrier

The glomerulus consists of four different cell types: parietal epithelial cells, podocytes (visceral epithelial cells), glomerular endothelial cells (GECs) and mesangial cells (Figure 1A). Parietal epithelial cells line the Bowman’s capsule, where the pre-urine is collected and forwarded to the proximal tubule. Mesangial cells are contractile cells that make up the mesangium and structurally support the glomerular tuft. Podocytes tightly wrap around and support the glomerular capillary vessels by an elaborate net of interdigitating foot processes. Between the foot processes are slit diaphragm proteins (e.g., nephrin and podocin), allowing contact between the podocytes and forming a size selectivity barrier for the passage of molecules and maintenance of glomerular filtration [13]. GECs cover the luminal surface of glomerular capillaries and are the cells of the glomerulus in direct contact with the blood. GECs and podocytes share a common extracellular matrix, the glomerular basement membrane (GBM), and together they form an interconnected glomerular filtration barrier.

Podocyte depletion associated with the progression of DKD has been extensively studied for mechanistic delineation in the breakdown of the glomerular filtration barrier [14]. Disease progression affects the intricate structure of the podocytes and leads to foot process effacement (FPE) [13]. There are several alterations in podocyte structure and function associated with FPE in DKD, including dedifferentiation (epithelial-to-mesenchymal transition), cytoskeletal rearrangement, impaired autophagy and apoptosis, which have been reviewed elsewhere [15,16,17,18]. Although podocytes have been studied extensively as primary targets in DKD, more recently, GEC dysfunction has been attributed to the pathogenesis of glomerular sclerotic diseases, including DKD [19,20,21,22].

The glomerular endothelium is among the unique vascular structures in the body. GECs are highly fenestrated, with fenestrae that are about 17 times larger than the diameter of albumin [23]. This structure allows for high water permeability, or hydraulic conductivity, needed for the large filtration volumes handled by the glomerulus [24]. The apical surface of the endothelial cells is covered by the negatively charged endothelial glycocalyx and endothelial surface layer (ESL), which cover and floats into the lumen of the capillary vessels, and is a key player in the integrity of the glomerular filtration barrier. The endothelial glycocalyx and ESL consist of glycoproteins and proteoglycans covalently linked to the glucosaminoglycans heparan sulfate, chondroitin sulfate and hyaluronic acid [25]. The importance of the glycocalyx in homeostasis has progressively been recognized, due in part to its high charge selectivity restricting the passage of negatively charged molecules such as albumin [23]. The presence of the glycocalyx also creates a space between blood and the endothelium and thereby controls vessel permeability, which leads to the regulation of water efflux [26]. The glycocalyx additionally restricts leukocyte and platelet adhesion to the endothelium, thus moderating inflammation and thrombosis, and allows an appropriate GEC response to flow variation through mechanosensing [26,27,28].

Endothelial cells are exposed to circulating high blood glucose levels and are particularly vulnerable to hyperglycemia-induced injury in diabetes; they then undergo a phenotypic switch that modifies their intracellular signaling leading to dysfunction [29]. Endothelial cell dysfunction is characterized by one or more of the following features: decreased nitric oxide (NO) bioavailability, reduced endothelium-mediated vasorelaxation, hemodynamic deregulation, impaired fibrinolytic ability, enhanced turnover and overproduction of growth factors. There can also be increased expression of adhesion molecules and inflammatory genes, excessive generation of reactive oxygen species (ROS), and enhanced permeability of the cell layer [30]. In diabetes, there is also evidence of a deficiency in endothelial progenitor cells involved in vascular regeneration [31]. Although different endothelial cell surfaces differ in the way they regulate glucose uptake, in GECs, hyperglycemia leads to saturation of glucose metabolism and results in activation of deleterious pathways such as the polyol pathway, the hexosamine pathway, the AGE/RAGE axis and the PKC pathway, leading to endogenous ROS overproduction [8,32]. In early DKD, activation of endothelial nitric oxide synthase (eNOS) that produces the vasodilator NO has been reported, while the progressive disease is associated with a deficiency of NO [33]. Reduced NO triggers uncoupling of eNOS in GECs and the formation of superoxide and peroxynitrite (ONOO^−^) [34,35]. This reaction can then oxidize the eNOS cofactor tetrahydrobiopterin (BH_4_), leading to more ROS and eNOS uncoupling [36]. Experimental models of T1DM and T2DM with eNOS knockout (eNOS^−/−^) have been shown to recapitulate glomerular lesions seen in human DKD, such as mesangial expansion, podocyte injury and depletion, albuminuria and focal segmental glomerular sclerosis (FSGS) [37,38]. These studies also suggest that podocytes, as well as mesangial cells, receive pathogenic signals from GECs in diabetes, indicating crosstalk among the cells in the glomerulus and interdependence.

## 3. Crosstalk between GECs and Podocytes Is Essential for Filtration Barrier Function and Is Disturbed in DKD

Balanced intercellular crosstalk between podocytes and endothelial cells is required for maintenance of the filtration barrier (Figure 1B). The best-studied mechanism of this crosstalk involves podocyte production of vascular endothelial growth factor-A (VEGFA), acting through paracrine signaling on VEGF receptors (VEGFR1 and VEGFR2) on GECs under physiological conditions. Eremina et al. elegantly demonstrated how conditional deletion of the VEGF gene in podocytes led to a loss of endothelial fenestrae, microangiopathy and proteinuria in otherwise healthy mice [39]. Others have shown that in VEGF-stimulated murine GECs, there is an increase in phosphorylation of eNOS, relating VEGF signaling to NO synthesis [40]. Today with the help of genetic models, we have a greater appreciation of the importance of VEGF-A imbalance in the diabetic setting. Studies using podocyte-specific VEGF-A ablation in STZ-diabetic mice resulted in the development of heavy proteinuria, marked glomerulosclerosis and glomerular cell apoptosis [41]. In contrast, earlier studies have suggested pharmacological inhibitors of VEGF activity to be beneficial in preventing kidney disease [42,43]. On the other hand, increased podocyte-derived VEGF-A was shown to be deleterious in non-diabetic mice, and the injury was further exacerbated with diabetes induction, resulting in advanced glomerulopathy with massive proteinuria [44]. Another example of podocyte-to-endothelial crosstalk is angiopoietins produced by podocytes, which are important for endothelial cell function [45]. They are critical for modulating the vascular response after the onset of diabetes and function as endothelial cell-protective factors in diabetes [46]. In addition, human biopsies, as well as experimental models of FSGS, have uncovered a causative role for the endothelin-1/endothelin receptor type A (Ednra) axis in promoting endothelial cell dysfunction and loss of the glomerular endothelial glycocalyx by increased degradation of glucosaminoglycans [47,48,49]. In diabetes, an increase in circulating as well as local production of endothelin-1 was demonstrated to activate Ednra in glomerular endothelial cells, resulting in mitochondrial stress and endothelial dysfunction [22] (Figure 1B). Podocytes or infiltrating macrophages may also promote the production of the heparan sulfate degrading enzyme heparanase, resulting in loss of glycocalyx [49,50].

When considering endothelial cells-derived intercellular signals, Isermann et al. have shown that the thrombomodulin-dependent formation of activated protein C by glomerular endothelial cells can support podocyte viability via protease-activated receptor1 (PAR-1) and the endothelial protein C receptor (EPCR) [51]. In diabetes, however, this pathway is disturbed and causes mitochondrial apoptosis pathway activation in podocytes and increased glomerular damage, leading to DKD [51]. GEC to podocyte crosstalk is regulated by shear stress, and studies have shown a critical pathway involving an ERK5 mediated increase in expression of KLF2 and downstream molecules, promoting an anti-coagulant, anti-inflammatory phenotype, which has been shown to directly affect podocyte function in co-cultures [52]. A study showed that endothelial cell knockout of KLF2 resulted in podocyte injury in diabetes and reduction of the endothelial glycocalyx [53]. More recently, the GEC and podocyte-specific contribution of TGF-β signaling in the progression of DKD has been examined; the study further highlights the importance of intricate crosstalk between injured glomerular cells [54]. Histological assessment of human biopsies from patients with T1DM and T2DM have unequivocally demonstrated podocyte as well as GEC injury [55,56]. Systematic assessment of glomerular capillaries using experimental models in early diabetes have provided insights suggesting that GEC injury may precede podocyte FP fusion [22] and may predispose to albuminuria either directly or indirectly by communication with neighboring podocytes via secreted mediators of exosomes [57,58]. Importantly, GEC injury and dysfunction were demonstrated to be absolutely necessary for subsequent podocyte depletion [22,47,48]. These studies support the importance of podocyte and GEC crosstalk in the maintenance, as well as the breakdown of the filtration barrier observed in DKD.

## 4. Oxidative Stress in DKD

A potential mechanism for GEC injury in DKD pathogenesis are reactive oxygen species (ROS), which play a direct role in diabetes and its complications, including nephropathy [59]. ROS are oxygen-derived, highly reactive molecules. Among them are free radicals such as superoxide (O_2_^−^) or hydroxyl (•OH) and non-radical ROS such as hydrogen peroxide (H_2_O_2_) [60]. ROS have a physiological role in cell signaling concerning cell proliferation and survival, which is under tight regulation and balanced by the cell’s antioxidant response [61]. Unless high levels of ROS are locally deployed for fighting against pathogens, excess ROS leads to oxidative stress, resulting in several cellular changes that can lead to organ dysfunction. Oxidative stress derived from the major sources of ROS has emerged as a causative mechanism in vascular dysregulation in disease states, including diabetes [59,62].

Metabolic changes (e.g., hyperglycemia, dyslipidemia) accompanying the diabetic pathology leads to increased circulation of noxious substances (e.g., glycated proteins and free fatty acids), a saturation of glucose metabolism pathways, and disturbed cellular redox balance. In the glomerulus, in particular, the diabetic milieu triggers oxidative stress responses in all cells via several endogenous pathways, including oxidative phosphorylation in mitochondria, NADPH oxidases (NOX), cytochrome P450, xanthine oxidase and uncoupled eNOS. Studies using increased extracellular glucose (30 mmol/L) found that it can rapidly stimulate intracellular ROS generation of conditionally immortalized podocytes via NADPH oxidase [63]. Another enzymatic source of extracellular and intracellular ROS in diabetes comes from increased xanthine oxidoreductase (XOR) activity [64]. An increase in mitochondrial ROS and mitochondrial dysfunction also plays a critical role in the pathogenesis of DKD [65]. There is now accumulating evidence that supports that the diverse sources of ROS, the timing, the location and the type of oxidative damage generated are important in the initiation and progression of kidney diseases. The next sections will focus on the major ROS generating pathways in DKD, such as NADPH oxidase, XOR and mitochondrial-derived ROS, as well as their potential interplay as it pertains to GECs injury in early DKD.

### 4.1. Active Enzymatic ROS Generation in DKD

#### 4.1.1. NADPH Oxidase (NOX)

NADPH Oxidases (NOX) have been suggested to contribute to the initiation and progression of DKD and other diabetic complications, and their activity is elevated significantly following the onset of hyperglycemia and increased circulating angiotensin II (Ang II), advanced glycation end products (AGEs) and TGFβ1 [66]. The biological function of NOX enzymes is to generate ROS by transferring electrons across biological membranes [67]. There are seven isoforms of NADPH oxidases, NOX 1-5 and dual oxidases (DUOX) 1 and 2. All isoforms catalyze the reduction of molecular oxygen (O_2_) to superoxide (O_2_^−^) using NADPH as an electron donor [60]. NOX4, however, has been shown to produce hydrogen peroxide (H_2_O_2_) rather than superoxide in vitro [68,69]. The regulatory machinery of the NOX isoforms differs. For example, ROS production by NOX 1–4, but not 5 requires binding to the subunit p22^phox^ [70]. NOX1 and NOX3 additionally bind the regulatory proteins NOX organizer 1, NOX activator 1 and Rac, although Rac appears more important for activation of NOX1 than of NOX3 [70]. NOX4, on the other hand, has high constitutive activity and may require only p22^phox^ for activation [60,70]. In contrast, regulation of NOX5, the most recently identified member of the NADPH oxidases, is the only NOX dependent on intracellular calcium [71,72,73,74]. The kidney has a distinct profile for NOX expression within the renal tubular cells, glomerular cells and in the vasculature [67,75].

There has been a significant research effort in the past decade focused on NOX function in the diabetic kidney. NOX4 has been of particular interest due to its enrichment in kidney tissue. In the glomerulus, NOX4 upregulation in glomerular mesangial cells in response to Ang II was shown to be associated with hypertrophy and fibronectin accumulation [76], and NOX4 together with NOX1 and CYP4A were shown by the same research group to mediate increased ROS production in podocytes in response to high glucose [77]. In contrast, other studies suggest that NOX4 expression levels were down rather than upregulated in tubular epithelial cells in chronic kidney disease (CKD) [78], as well as other reports showing a protective effect of NOX4 in the vasculature after ischemia-induced or inflammatory injury [79]. A study by Zhao et al. showed that hyperglycemia induced the upregulation of hedgehog interacting protein in GECs, which stimulated fibrosis through TGFβ-signaling, inducing apoptosis of GECs by NOX4 generation of H_2_O_2_ [80]. These studies in aggregate suggest that the timing and localization of ROS production by NOX4 are important in determining the effect of NOX4 enzymatic activity. Interestingly, overexpression of human NOX2 in endothelial cells of Akita T1DM mice led to increased superoxide production, decreased thickness of the endothelial glycocalyx, mesangial matrix expansion and increased podocyte damage with proteinuria [81], despite the mice having a C57BL/6 background; known to be a relatively resistant strain to development of DKD [82]. Importantly, a recent study by the Jandeleit-Dahm research group presented evidence for deleterious effects of endothelial NOX5 expression resulting in increased ROS production in non-diabetic conditions, which was exacerbated by diabetes [83]. These studies suggest that GEC specific NOX-derived ROS can result in glomerular injury in DKD.

Considering the evidence for increased NOX activity and associated ROS production in DKD, NOX’s have been explored as potential therapeutic targets. The dual specific NOX1/NOX4 inhibitor GKT137831 was successful in ameliorating glomerular structural changes, albuminuria and fibrotic signaling in diabetic mice models [84,85]. However, in humans, a phase II clinical trial did not show a reduction of proteinuria compared to placebo [86]. GKT137831 has since been tested as a regulator of fibrosis in the autoimmune liver disease primary biliary cholangitis with promising results [87]. The pan NOX inhibitor APX115 has also demonstrated renal protective effects in preclinical studies involving a T2DM murine model [88], and a clinical trial is currently underway with T2DM patients [89].

#### 4.1.2. Xanthine Oxidoreductase (XOR)

Xanthine oxidoreductases (XOR) are xanthine dehydrogenase (XDH) and xanthine oxidase (XO), interchangeable forms of the same enzyme encoded by the *XDH* gene. They catalyze the oxidation of purine substrates, xanthine and hypoxanthine and use NAD^+^ as an electron acceptor. In humans, the enzymatic oxidation of hypoxanthine to xanthine and further to uric acid by XO uses O_2_ as an electron acceptor and generates H_2_O_2_ and O_2_^−^ [90,91]. Uric acid (UA) is the endpoint of purine metabolism in humans, and both hyperuricemia and hypouricemia can have negative consequences for renal health [91,92].

High XOR activity was shown to be correlated with high serum UA levels, as well as with insulin resistance, adiposity, and subclinical inflammation [93], and was an independent predictor of diabetic complications among T2DM patients [94]. Importantly, increased XOR in circulation is strongly associated with ESRD [95] and have been shown to be risk factors for cardiovascular diseases and DKD [96,97]. Although the role of UA as a risk factor for CKD has been largely debated, there are many studies supporting its role in the development and progression of kidney fibrosis, vascular dysfunction, as well as the benefits of XOR inhibitors in these conditions [96,97,98,99,100,101,102,103,104].

Prospective studies involving patients with type 1 diabetes have shown that higher serum urate levels are associated with an increased risk of rapid GFR decline [105,106]. Hence, efforts to decrease serum UA have been assessed for efficacy in CKD. Outcomes from two recent high profile trials; the Preventing Early Renal Loss in Diabetes (PERL) trial and the Controlled Trial of Slowing of Kidney Disease Progression from the Inhibition of Xanthine Oxidase (CKD-FIX) over 3 and 2 years, respectively, did not show benefit in T1DM patients with mild to moderate kidney disease [107,108]. Although the hypothesis tested was to lower UA, effects on XOR-derived ROS were not examined, despite the many reports supporting that the tested substance allopurinol causes ROS through self-oxidation to form oxypurinol, resulting in the reduction of O_2_ [109,110]. This initial reaction drives substrates such as xanthine to donate electrons with enzyme turnover reactions that result in excess ROS formation before inhibition of UA is attained [111]. This undesirable action of allo/oxypurinol ROS generation has led to significant misinterpretation of ROS-driven pathology where XOR is a contributor. Febuxostat, a non-purine XOR inhibitor [112], was shown in experimental DKD to decrease ROS damage. The anti-albuminuric and the renoprotective effects observed were shown to be attributed to attenuation of the inflammatory and oxidative stress [113]. Febuxostat was able to slow the decline in eGFR in CKD stages three and four compared to placebo in a smaller trial [114], with no adverse events observed [115]. These studies and trials highlight the need to understand the chemistries of XOR inhibitors and for future studies to focus on ROS production in patients.

#### 4.1.3. Mitochondrial ROS

Mitochondria are the energy-producing organelles in cells via the generation of ATP through oxidative phosphorylation. The kidney consumes a large amount of energy for the reabsorption of large quantities of fluid and solutes across the renal tubular epithelium. Hence it is not so surprising that renal disease can be observed in inherited mitochondrial disorders, including Kearns–Sayre syndrome, Pearson syndrome, DIDMOAD (Wolfram’s syndrome), and Leigh syndrome [116]. Mitochondrial mutations have been associated with childhood-onset FSGS and steroid-resistant nephrotic syndrome [117,118,119,120], and mitochondrial function seems to be crucial for the maintenance of the glomerular filtration barrier [121]. This is even though podocytes under physiological conditions rely on anaerobic glycolysis as the predominant energy source [122]. Endothelial cells similarly generate >75% of their ATP via glycolysis, despite abundant access to oxygen [123,124]. Mutations in genes involved in coenzyme Q10 biosynthesis in podocytes, important in supporting electron transport of oxidative phosphorylation, or in complex IV assembly cofactor heme A: farnesyltransferase in cells of the developing nephrons, were sufficient to cause FSGS [125,126,127], underscoring the importance of mitochondria for and beyond energy production.

Mitochondria conduct other key cellular functions, such as homeostasis of calcium and iron, regulation of tissue oxygen gradients, H_2_O_2_ signaling and fatty acid uptake [128,129], as well as biosynthesis of heme, pyrimidines, steroids and modulation of programmed cell death [130,131]. ROS are byproducts from the oxidative phosphorylation reaction, and a significant portion of electrons (0.2% of the oxygen consumed) normally escape the electron transport chain as superoxide anions (O_2_^−^). This figure can increase to up to 2% under conditions of oxidative stress [132,133,134], resulting in damage to mitochondria and activating a vicious cycle of more ROS generation, eventually resulting in loss of cell function and tissue abnormality. The deleterious impact of excess mitochondrial ROS, together with a decrease in antioxidative defense systems, are involved in the pathophysiology of DKD, affecting both the glomerulus and the tubular system [135,136,137].

Although there are relatively few studies exploring the effects of mitochondrial dysfunction and ROS production specifically in glomerular endothelial cells [138], research from our lab has shown that genes involved in oxidative phosphorylation and mitochondrial dysfunction were the most enriched in a transcriptomic comparison of mice susceptible and resistant to DKD [22]. Importantly, mitochondrial oxidative stress and DNA damage in DKD susceptible mice was specific to endothelial cells, resulting in loss of endothelial fenestrae and subsequent podocyte depletion [22]. Other studies have shown that the mRNA profiles of isolated GECs and podocytes from diabetic mice kidneys demonstrated distinct upregulated pathways involving mitochondrial function and oxidative stress in the endothelium compartment. Meanwhile, changes in the regulation of actin cytoskeleton-related genes were the major pathways affected in podocytes isolated from diabetic mice [139]. Furthermore, deleterious effects on the endothelial glycocalyx associated with increased mitochondrial ROS exposure contribute to the breakdown of the glomerular filtration barrier [48,140]. More recently, we have demonstrated that diabetic milieu-mediated GEC mitochondrial oxidative stress and impaired autophagy resulted in oxidative damage accumulation in vitro, while exposure of podocytes to the same diabetic milieu resulted in minimal oxidative stress [141]. Interestingly, factors secreted by the stressed GECs caused podocyte apoptosis, while the effect was blocked by the addition of the mitochondrial ROS scavenger MitoTEMPO [141]. These findings provide evidence of endothelial cell mitochondrial dysfunction and overproduction of ROS as early insults can trigger podocyte injury and, therefore, breakdown of glomerular filtration barrier through intercellular crosstalk in DKD. Importantly, cell-specific ROS overproduction could have cells specific distinct and important roles in the glomerulus.

Much remains to be elucidated in defining the functional role of mitochondria in DKD. However, restoration of mitochondrial function could be beneficial. Some mitochondrial interventions currently being explored include the Szeto–Schiller peptide elamipretide (MTP-131), a tetrapeptide that targets mitochondrial cardiolipin, which demonstrated benefits in rodent models of DKD by improving mitochondrial bioenergetics [142,143] and protected mitochondrial cristae structure in both GECs and podocytes [135,144]. The peptide is currently being evaluated for cardiac and renal effects in hospitalized heart failure patients. The efficacy of coenzyme Q10 supplementation was reported to be promising for the treatment of DKD [145] and shown to improve mitochondrial function and decrease oxidative stress in patients receiving hemodialysis [146]. Due to coenzyme Q10 being lipophilic in nature, transport to the mitochondrial inner-membrane is limited. Thus a more hydrophilic intermediate such as 2,4-dihydroxybenzoic acid, which is found naturally in certain foods, can reactivate coenzyme Q10 levels [147,148] and could be of benefit to DKD patients. The mitochondrial-targeted ROS scavenger MitoQ, a form of coenzyme Q with a lipophilic cation for enrichment in mitochondria, have been shown to convey renoprotective effects, with improved albuminuria and hyperfiltration, but not hypertrophy and mesangial expansion, in T2DM mice [149]. MitoQ is currently being evaluated clinically to examine microvascular function in patients with moderate to severe CKD. Finally, boosting antioxidant and mitochondrial biogenesis pathways by activation of the transcription factor, nuclear factor erythroid-2 related factor 2 (NRF-2), has been shown to improve kidney function in a number of glomerular diseases, although there was no reduction in proteinuria [150]. Clinical trials are underway to evaluate the cardiac and renal benefits of mitochondria stabilizing agents in patients. The question of whether or not mitochondrial stabilization and mitochondrial ROS inhibition can improve patient endpoints in large, randomized DKD clinical trials still remains.

### 4.2. ROS Interplay

In diabetes, oxidative stress may result from an interplay between different ROS sources resulting in a vicious cycle in glomerular cells, and as discussed above, this can result in impaired GEC function (Figure 2). “Redox switches” have been identified in different sources of superoxide, hydrogen peroxide, and peroxynitrite, for example, for the conversion of XDH to the XO form or for the uncoupling process of eNOS [151]. Both ROS and UA are products of XOR reaction and have been shown to induce mitochondrial dysfunction and reduced mitochondrial mass and ATP production in diabetes [100,152]. XOR products can also downregulate mitochondrial metabolism by increasing mitochondrial calcium and stimulating superoxide production [104]. Mitochondrial permeability transition pore is affected by ROS produced by non-mitochondrial sources and result in an increase in peroxynitrite (ONOO−) with eNOS uncoupling, as well as mitochondrial protein, RNA and DNA damage [153,154]. In addition, mitochondrial ROS scavengers can influence XO activity, as demonstrated by the improvement of cardiac complications and XO activation with MitoQ [155]. Moreover, increased angiotensin-II-dependent NADPH oxidase activation in diabetes can mediate mitochondrial dysfunction with subsequent mitochondrial-derived ROS formation [156]. Importantly, the Nox4 isoform was reported to be localized in mitochondria in diabetes and could contribute to processes that are associated with mitochondrial oxidative stress [157]. However, to this date, there is only limited evidence for redox-based activation pathways of NOX, XO and for the role of mitochondrial ROS in DKD. Understanding these interactions is important, as not all ROS are the same. ROS are produced both under physiological and pathological conditions. Hence general antioxidant therapy approaches have failed in large clinical trials with DKD patients [158,159]. More research is needed to further the general understanding of the contribution of redox processes in DKD.

### 4.3. Current Clinical Approaches for DKD and Their Effects on ROS

In addition to the pharmacological agents targeting the different sources of ROS described above, there are ongoing investigations of other primary pharmaceutical targets that impact ROS in DKD. For instance, the selective endothelin 1 receptor A (Ednra) antagonist atrasentan has been demonstrated to have beneficial effects on renal function and proteinuria in T2DM patients in phase 3 randomized clinical trial SONAR [160]. Despite adverse effects, including fluid retention and heart failure accompanying Ednra inhibition in DKD [161], encouraging outcomes from the phase 2 DUET trial in people with primary FSGS with sparsentan combined with angiotensin receptor blocker [162] are being further explored in the Phase 3 DUPLEX trial. The proven antiproteinuric effects of Ednra blockade could be attributed to the prevention of pathologic crosstalk between podocytes and GECs in DKD, where increased Ednra signaling in GECs leads to mitochondrial oxidative stress and damage of GECs [22]. As previously discussed, GEC stress and dysfunction mediated podocyte injury and depletion [47], and inhibition of Ednra was found to be beneficial in maintaining the endothelial glycocalyx [48,49,50]. Another mechanism that indirectly ameliorates oxidative stress and is being explored in patients with DKD is the selective induction of ATP-binding cassette A1 (ABCA1) [163,164]. ABCA1 inducers promote the removal of excess cholesterol from podocytes and therefore stabilizes mitochondrial cardiolipin in podocytes in DKD [163].

Exciting outcomes from recent trials testing the effectiveness of sodium–glucose cotransporter 2 (SGLT2) inhibitors have produced great expectations with positive effects on hyperglycemia control, as well as on cardiovascular and renal outcomes of T2DM. SGLT2 inhibitors have systematic pleiotropic effects, including the normalization of altered tubuloglomerular feedback, and hence can control diabetes-induced hyperfiltration and intraglomerular pressure. Interestingly, other effects of SGLT2 inhibitors that are independent of blood glucose control or bodyweight reduction are the amelioration of mitochondrial damage in tubular cells [165] as well as reduction of UA [166]. In experimental DKD, these agents reduced mesangial expansion, accumulation of extracellular matrix proteins, and podocyte injury [167]. A study showed that canagliflozin inhibited high-glucose-induced activation of the protein kinase C-NOX pathway and ROS production in cultured mesangial cells [168]. Although no studies have reported the effect of SGLT2 inhibitors on ROS in GECs, empagliflozin and dapagliflozin were shown to restore barrier function, adhesion molecules expression, NO bioavailability and inhibit ROS production in endothelial cells under inflammatory conditions [169], and these effects were independent of SGLT2 expression in these cells. A recent study also suggested that the glucagon-like peptide 1 agonist liraglutide can ameliorate uncoupling of the VEGF-NO axis by activation of the AMPK–eNOS pathway in glomeruli of mice with obesity-related kidney disease [170]. Studies assessing their effects on GECs are expected to provide valuable tools in DKD therapy in the upcoming years.

## 5. Conclusions

The diabetic state plays a pivotal role in the stimulation of excess ROS generation in the kidney. Increased ROS production and accumulation of ROS damage products promote injury in all glomerular cells, especially in GECs, resulting in dysfunction. Impaired endothelial function in DKD is now recognized to cause functional and structural changes within the glomerulus through crosstalk [20,138,171], and this may be central to early pathogenesis and help drive disease progression. Though being generated from many sources in diabetes, ROS from NADPH oxidase, XORs and mitochondria are thought to cause the onset of albuminuria followed by a progression of renal damage. Still, the validation of findings in experimental models and translation to humans remains a point of controversy. Better understanding of the context which results in ROS overproduction, timing, and cell–cell interactions may lead to greater insights regarding the potentially reversible events that lead to DKD progression and inspire novel therapeutic approaches.

## Figures and Tables

**Figure 1 ijms-21-09456-f001:**
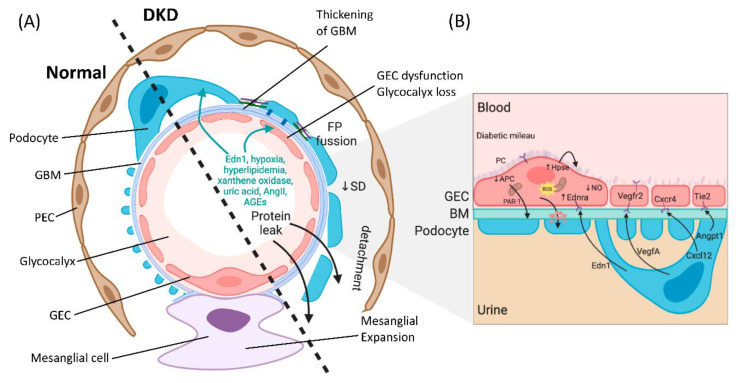
The glomerular filtration barrier. (**A**) Schematic showing a glomerular capillary loop that forms the primary filtering unit of the kidney, illustrating normal (**left**) and diabetic kidney disease (DKD) (**right**) morphology features. The glomerular capillary loop is comprised of specialized structures: parietal epithelial cells (PECs), the mesangial cells, visceral epithelial cells (podocytes) with interdigitating foot processes (FP) attached to the glomerular basement membrane (GBM), fenestrated glomerular endothelial cells (GEC), and the glycocalyx. Between podocyte foot processes is the slit diaphragm (SD), forming a bridge and tethering foot processes together. In DKD, there are profound morphological changes to the glomerulus, including thickening of the GBM, mesangial matrix expansion and increases in extracellular matrix deposition occluding its capillaries. Additional changes are GEC dysfunction, degradation of the glycocalyx, fusion of foot processes, disrupted architecture of the slit diaphragm, podocytes depletion and consequential protein leak. (**B**) There is disrupted bidirectional signaling among the cells in the filtration barrier in DKD, including increased podocyte-derived vascular endothelial growth factor a (VEGFA) in early DKD and decrease of VEGF in progressive disease. There is a loss of angiopoietin-1 (Angpt1) and tyrosine-protein kinase receptor (Tie2) interaction in diabetes, and production of activated protein C (APC) in the glomerulus is reduced because of suppression of thrombomodulin expression. Decreased functional activity of APC affects the permeability of the glomerular capillary wall and enhances apoptosis of GECs and podocytes. Endothelin-1 (Edn1) and endothelin receptor A (Ednra) pathway activation in DKD lead to mitochondrial oxidative stress, endothelial nitric oxide (NO) depletion, and degradation of the endothelial glycocalyx.

**Figure 2 ijms-21-09456-f002:**
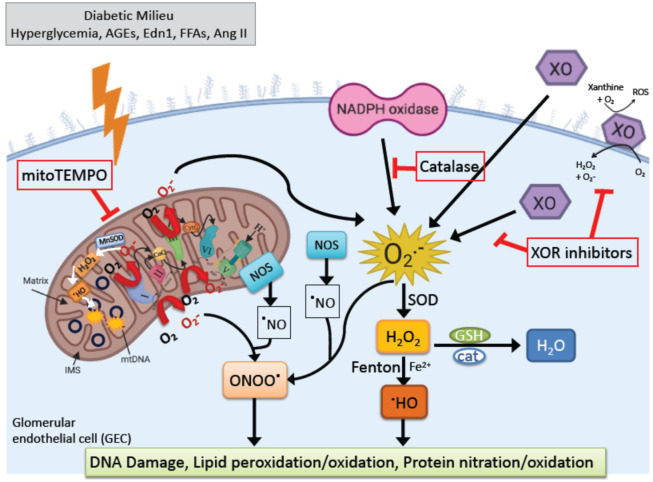
Generation of excess reactive oxygen species (ROS) in the GECs. Pathological metabolic conditions associated with diabetes mellitus (including hyperglycemia, increased circulation of advanced glycation end products (AGEs), endothelin-1 (Edn1) and free fatty acids (FFAs), as well as activation of RAAS leading to release of angiotensin II (Ang II)) can exacerbate ROS production over the cell’s antioxidant capacity, and this imbalance contributes to endothelial dysfunction in DKD. ROS are generated by enzymatic and nonenzymatic redox reactions during cellular metabolism under normal and pathological conditions. The superoxide anion (O_2_•^−^), generated in mitochondria, plasma membrane, peroxisomes, and cytosol, becomes the precursor free radical for the generation of other ROS molecules. Next, cytosolic CuZN superoxide dismutase (SOD) and mitochondrial MnSOD convert O_2_•^−^ to H_2_O_2_, which yields highly reactive hydroxyl radicals (OH•) by interaction with reduced transition metal ions (such as Fe and Cu) in a Fenton reaction. In addition to ROS, cells also generate reactive nitrogen species (RNS). The major RNS include nitric oxide (•NO), peroxynitrite (ONOO^−^), and nitrogen dioxide (•NO_2_). Nitric oxide (•NO) is produced by three isoforms of nitric oxide synthase (NOS). Finally, the excess ROS produced causes oxidative damage to mitochondrial and nuclear DNA, lipid and protein oxidation, protein nitration, and mitochondrial dysfunction.

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
