# Peer review of "Molecular Mechanisms in Early Diabetic Kidney Disease: Glomerular Endothelial Cell Dysfunction"

_ijms, 2020, doi:10.3390/ijms21249456_

Round 1

Reviewer 1 Report

Daechn and Lassen presented an interesting study: Molecular mechanisms in early DKD: glomerular endothelial cell dysfunction.

The authors addressed an interested topic, presented relevant previously published studies, correctly stressed and discussed the results and some limitations of the presented studies, highlighted current therapeutic approach and made some interesting conclusions.

Author Response

We appreciate the Reviewer's comments. We enjoyed discussing this topic and are pleased that the Reviewer found it interesting.

Reviewer 2 Report

Lassén and Daehn reviewed the molecular mechanisms in early DKD and shed on light the major contribution of glomerular endothelial cell dysfunction.

The authors first present the glomerular structure with appropriate figure to guide the readership into this complex topic

The authors  elegantly discussed the crosstalk between glomerular endothelial cells and podocytes, with references to in vitro and in vivo studies. The role of oxidative stress is emphasized with figures and reference to recent literature.

This is a well written and comprehensive review with didactic and helpful figures for the readership. 

Interestingly, beyond this excellent scientific presentation,  the authors point out the major economic burden of diabetes complications and the therapeutic relevance of the underlying molecular mechanisms described.

Minor comments

May the authors spell out DKD in the title?

May the authors clarify the structure of the review ? Notably the section 2 is subdivided in "2.1 Crosstalk between GECs and podocytes is essential for filtration barrier function and is disturbed in DKD" but there is no "2.2"

Author Response

We appreciate the Reviewer's comments.

According to the reviewer’s suggestion, we have made the changes to the title and the structure of the paper. We have also made spelling and grammar corrections though out the manuscript to improve readability. 

Reviewer 3 Report

This review article is well-written and understandable.

Readers whose major is not nephrology can understand the pathophysiology of diabetic kidney disease and chronic kidney disease.

Author Response

We are grateful for the Reviewer's encouraging comments.